# Paradoxes in Fair Machine Learning

**Paul Gölz, Anson Kahng, and Ariel D. Procaccia**
Computer Science Department
Carnegie Mellon University
{pgoelz, akahng, arielpro}@cs.cmu.edu

## Abstract

Equalized odds is a statistical notion of fairness in machine learning that ensures that classification algorithms do not discriminate against protected groups. We extend equalized odds to the setting of cardinality-constrained fair classification, where we have a bounded amount of a resource to distribute. This setting coincides with classic fair division problems, which allows us to apply concepts from that literature in parallel to equalized odds. In particular, we consider the axioms of resource monotonicity, consistency, and population monotonicity, all three of which relate different allocation instances to prevent paradoxes. Using a geometric characterization of equalized odds, we examine the compatibility of equalized odds with these axioms. We empirically evaluate the cost of allocation rules that satisfy both equalized odds and axioms of fair division on a dataset of FICO credit scores.

## 1 Introduction

Throughout most of human history, the question "who deserves what?" could only be answered by people. As such, questions of fairly allocating resources among groups of people were historically dictated by common sense, enforced by law, or suggested by social conventions. In the age of big data, however, machine learning algorithms increasingly dictate decisions about distributing resources in a wide range of domains [15, 19]. Machine learning classifiers have been trained to determine which applicants deserve bank loans [19], which students merit acceptance from a particular school [23], or which prisoners should receive parole [15]. The prevalence of algorithmic intervention has led to a widespread call for accountability in machine learning: in order to ensure that algorithms do not disproportionately affect different constituent subpopulations, researchers must be able to provide fairness guarantees of the resulting classification algorithms. This call, in turn, has led to much prior work on measuring and ensuring statistical notions of fairness, notably through metrics like demographic parity and equalized odds [8, 10, 11, 13, 16, 18, 24–26].

The statistical notion of fairness that we will consider throughout this paper is that of *equalized odds*, which states that a classifier must have equal true positive and false positive rates for all groups. While equalized odds has been extensively studied as a metric of fairness in machine learning [10, 11, 13, 16, 18, 24], it has not been considered in settings where a desired number of positive labels is given. This constraint is natural and ubiquitous whenever agents labeled as positive obtain a limited resource. For instance, a school can only offer admission to a fixed number of students, a police department's staff dictates the number of suspects they can stop and frisk, and a bank might only have a finite amount of available loans. In the unconstrained setting, the quality of a classifier is computed by adding a given utility per true positive and subtracting a given cost per false positive. In the cardinality-constrained setting, the efficiency that we seek to maximize is simply the number of true positives (e.g., people who repay loans or students who will graduate from school). Since we fix the number of overall positives, optimizing for any choice of (positive) utility and (positive) cost coincides with maximizing our notion of efficiency.

While fair classification has not previously been studied from this perspective, the task of fairly allocating a finite resource is central to the field of *fair division* [6, 17]. Indeed, it is natural to directly formulate the problem of fairly classifying agents, where exactly $k$ must be labeled as positive, as the fair division problem of awarding $k$ identical items to $k$ applicants in a way that satisfies certain fairness constraints.

That being said, the notions of fairness differ between the fairness in machine learning and fair division communities. On one side, the machine learning literature studies statistical notions of fairness that hold over groups and which are usually mutually exclusive. In contrast, the fair division literature includes a whole toolbox of fairness axioms, most of which can be understood as precluding a paradox that, if present, would clearly violate intuitive notions of fairness. Combinations of these axioms then induce families of allocation rules that are immune to these types of paradoxes. To the best of our knowledge, there has been no prior work that relates statistical measures of fairness to classical axioms of fairness. This is unfortunate, since it would certainly be desirable to prevent the corresponding types of paradoxes when applying fair machine learning. This motivates our main research question: *To what extent is equalized odds compatible with axioms of fairness prevalent in fair division?*

Our contributions are twofold: First, we introduce the setting of cardinality constraints and study optimal classification algorithms that satisfy equalized odds in this setting. In particular, we present a geometric characterization of the optimal allocation rule that satisfies equalized odds given cardinality constraints.

Second, in the cardinality-constrained model, we examine the relationship between equalized odds and the following three standard fair-division axioms. *Resource monotonicity* captures the intuition that, given more of a resource to distribute among a population, no agent should be worse off than before. *Consistency* says that, if an agent leaves with her allocation, then running the same allocation rule on the remaining agents and resources should result in the remaining agents receiving the same allocations as before. *Population monotonicity* states that, if an agent joins the division process, then all previous agents should receive at most what they previously received.

For resource monotonicity, we achieve a positive result: resource monotonicity can be implemented alongside equalized odds without cost to efficiency, which requires careful consideration of how goods are allocated inside of each group. For consistency, we prove a strikingly negative result — the only allocation rule that satisfies equalized odds and consistency is uniform allocation. In the case of population monotonicity, compatibility with equalized odds is also severely limited. More precisely, no allocation rule that achieves a constant approximation of the optimal equalized-odds efficiency can satisfy population monotonicity. To complement these theoretical results, we use a dataset of FICO credit scores to study the efficiency of allocation rules that satisfy equalized odds and each of the three axioms.

Our results are related to, but conceptually and technically distinct from, previous work showing that equalized odds is incompatible with other statistical notions of fairness, notably the property of calibration. Intuitively, calibration states that if a classifier assigns a probability label of $p$ to a set of $n$ people, then $p\,n$ of them should actually be positive [10, 16, 18]. It has been shown [16, 18] that when groups have different base rates, i.e., probabilities that they belong to the positive class, the only classifier that satisfies equalized odds and calibration is the perfect classifier. Note that our approach is not in conflict with these results; we assume a calibrated, unfair classifier and produce a fair, but uncalibrated classifier. Indeed, our final classifier should not be expected to be calibrated since the sum of allocations is determined by the cardinality constraint, not by the fraction of positive agents in the population. Additionally, work by Corbett-Davies et al. [11] establishes a trade-off between achieving equalized odds and the natural fairness notion of holding all agents in all groups to the same standard.

## 2 Our Model

We consider settings with at least two groups, and let $g$ range over these groups by default. Each group is composed of *positive* and *negative* agents; allocating a good to a positive agent is preferable to allocating it to a negative agent. For example, if we distribute loans, positive agents might be those who will not default if they are given the loan, or, if we select whom to stop and frisk, positive individuals might be those who indeed carry an illegal weapon. To exclude trivial cases, we assume that some positive and negative agents exist, even if not necessarily in the same group.

We assume the existence of a calibrated classifier on each group. Thus, for every group $g$, there is a finite set $P_g$ of probabilities of, say, repaying a loan. When simultaneously ranging over $g$ and $p$, we implicitly only refer to $p \in P_g$. For each $p \in P_g$, $d_g^p > 0$ gives the number of agents to whom the classifier assigns probability $p$ of being positive. We refer to the set of agents in the same group classified as the same probability as being in one *bucket*. By the calibration assumption, $p\, d_g^p$ agents in a bucket $(g, p)$ are positive and $(1 - p)\, d_g^p$ are negative. Denote the total number of positive agents in $g$ by $D_g^+ := \sum_{p \in P_g} p\, d_g^p$ and the total number of negative agents by $D_g^- := \sum_{p \in P_g} (1 - p)\, d_g^p$. The total cardinality of a group is $D_g := \sum_{p \in P_g} d_g^p$, and the total cardinality over all groups is $D := \sum_g D_g$.

## 2.1 From Classification with Cardinality Constraints to Allocation of Divisible Goods

An allocation algorithm is given the output of the classifier and a real-number cardinality constraint $k \in [0, D]$. The algorithm must allocate $k$ units of a divisible good to the agents, where each agent can receive at most one unit of the good. The objective in this allocation is to maximize *efficiency*, i.e., the amount of goods allocated to positive agents.

Note that this setting, where we allocate a divisible good, generalizes binary classification with cardinality constraints. Indeed, the latter problem is equivalent to distributing $k$ discrete indivisible items. If we choose our good as the probability of receiving an item, we can immediately apply our framework to this setting. Using the Birkhoff-von Neumann theorem [5, 22], the individual allocation probabilities can be translated into a lottery of the items that guarantees that exactly $k$ many items are distributed at any time.

That said, the increased expressive power does allow us to capture additional settings of interest. For example, in the context of the fair allocation of financial aid, colleges typically provide different amounts of aid to different students, rather than making binary decisions.

Returning to our model, for each bucket $(g, p)$, let $\ell_g^p \in [0, d_g^p]$ denote the amount of goods allocated to the agents in the bucket. Since the algorithm does not possess more detailed information than the classifier output, we may without loss of generality assume that the allocation equally spreads a bucket's allocation between its members. Indeed, if $0 < p < 1$, any unbalanced allocation inside the bucket would make mean allocations in the definition of equalized odds depend on which agents will be positive, which means that equalized odds cannot be guaranteed. For the probabilities $0$ and $1$, all agents in the bucket have the same type, and the algorithm can, in principle, arbitrarily discriminate between them. However, since the agents in the bucket are indistinguishable, assuming a balanced allocation does not change our analyses.

With these observations, we know that the total allocation to positive agents in group $g$ is $L_g^+ := \sum_{p \in P_g} p\, \ell_g^p$ and that the total allocation to negative agents is $L_g^- := \sum_{p \in P_g} (1 - p)\, \ell_g^p$. Let the cardinality of the group allocation be $L_g := L_g^+ + L_g^- = \sum_{p \in P_g} \ell_g^p$.

Each allocation is decomposable into allocations for each group. For a group $g$, we call a group allocation $(\ell_g^p)_p$ *uniform* if $\ell_g^p = \alpha\, d_g^p$ for some $\alpha \in [0, 1]$ and all $p \in P_g$. Another important class of group allocations are *threshold* allocations, which do not give any goods to agents in a bucket $p$ until every agent in a higher-$p$ bucket of the same group receives a full unit of the good. Formally, there must be a threshold probability $p^*$ such that $\ell_g^p = d_g^p$ for all $p > p^*$ and such that $\ell_g^p = 0$ for all $p < p^*$, where $\ell_g^{p^*}$ can be arbitrary.

## 2.2 Equalized Odds

Throughout the paper, allocations must satisfy *equalized odds*, which means that (a) the mean allocation over the positive agents in $g$ is equal between all groups $g$ that have any positive agents; and (b) the mean allocation over the negative agents in $g$ is equal between all groups $g$ that have any negative agents. We refer to the pair $(L_g^+/D_g^+, L_g^-/D_g^-)$ — the mean allocation to positive agents and the mean allocation to negative agents — as the *signature* of the allocation.

## 2.3 Fair-Division Axioms for Allocation Algorithms

There have been many decades of work on fair division, spanning settings with both divisible and indivisible goods [6, 7, 12, 14, 17, 20]. Throughout this literature, desirable properties are

encoded via axioms, which can be either *punctual* or *relational*. Punctual axioms such as equitability, proportionality [20], and envy-freeness [12] apply to each instance of the fair division problem separately, and ensure that each agent's allocation satisfies global or relative valuations. For example, proportionality states that, given $n$ agents, each agent should receive at least a $1/n$ fraction of her value for the entire resource, and envy-freeness states that all agents should value their own allocations more than any other allocation given to another agent [21]. By contrast, relational axioms such as resource monotonicity, consistency, and population monotonicity link separate instances of the fair division problem together and can be thought of as well-behavedness properties. In our model, each agent desires as much of the good as possible, which means that only uniform allocation satisfies punctual axioms such as equitability, proportionality, and envy-freeness.[1] For this reason, we focus on the abovementioned relational axioms.

An allocation algorithm satisfies *resource monotonicity* if increasing $k$ does not decrease any agent's allocation. In our model, we assume that the amount of goods to be allocated is fixed a priori. In practice, however, additional resources might become available during the allocation phase. If the availability of more resources were to decrease an agent's allocation, the allocator might find it difficult to recuperate goods that have already been promised (or distributed). Resource monotonicity avoids these bad situations.

*Consistency* says that allocations can be computed separately for subsets, using the share of the good allocated to them. Formally, for a given classification $(\hat{d}_g^p)_{g,p}$ and allocation cardinality $\hat{k}$, let $(\hat{\ell}_g^p)_{g,p}$ define the allocations of an allocation algorithm. Consider a second instance, in which we remove some agents, i.e., have an allocation $(d_g^p)_{g,p}$ such that $d_g^p \leq \hat{d}_g^p$ for all buckets (buckets might also be removed, which we represent by setting $d_g^p = 0$). In addition, we reduce the allocation cardinality to what these agents together received in the previous instance, i.e., have a new allocation cardinality $k := \sum_{g,p} d_g^p/\hat{d}_g^p \, \hat{\ell}_g^p$. Consistency requires that every agent of the second instance receive the same allocation as in the first, i.e., that $\ell_g^p = d_g^p/\hat{d}_g^p \, \hat{\ell}_g^p$. Notably, assuming both consistency and equalized odds implies that equalized odds must hold over subpopulations. For instance, fairness between racial groups would be preserved when considering only the female, senior, or foreign-born subpopulations; ruling out fairness analogues of Simpson's paradox. While this would certainly be desirable, we will show that it comes at an unreasonable price in efficiency.

Finally, *population monotonicity* mandates that, if we remove some of the agents without changing the allocation cardinality, the allocation to any remaining agent cannot decrease. In our example of allocating financial aid, for instance, it is quite likely that students will join another school or drop out after enrollment. If we want to preserve equalized odds, and if our allocation rule violates population monotonicity, the departure of a student from the application pool might reduce another student's allocation, which will be hard to justify.

Note that consistency and resource monotonicity together imply population monotonicity. Indeed, if we remove some agents together with their allocation, the allocation to the remaining agents does not change by consistency. Adding the removed goods back can only increase allocations by resource monotonicity.

## 3   Geometric Interpretation of Equalized Odds

As observed by Hardt et al. [13], the axiom of equalized odds is most easily understood through the lens of a geometric interpretation. We adapt and extend their interpretation to our setting and prove that it encompasses all equalized-odds allocations (which Hardt et al. do not do). The resulting characterization is employed to prove our axiomatic theorems in Section 4, and gives an algorithm used in Section 5.

For the time being, focus on a single group $g$ and ignore the cardinality constraint. An allocation to this group $(\ell_g^p)_p$ is now only constrained by $0 \leq \ell_g^p \leq d_g^p$ for all $p$.

Let $f_g$ be a function mapping every group allocation to its signature $(L_g^+/D_g^+, L_g^-/D_g^-)$ in $[0,1]^2$. Denote the image of $f_g$ by $S_g$, which marks the set of implementable signatures.[2] For an example

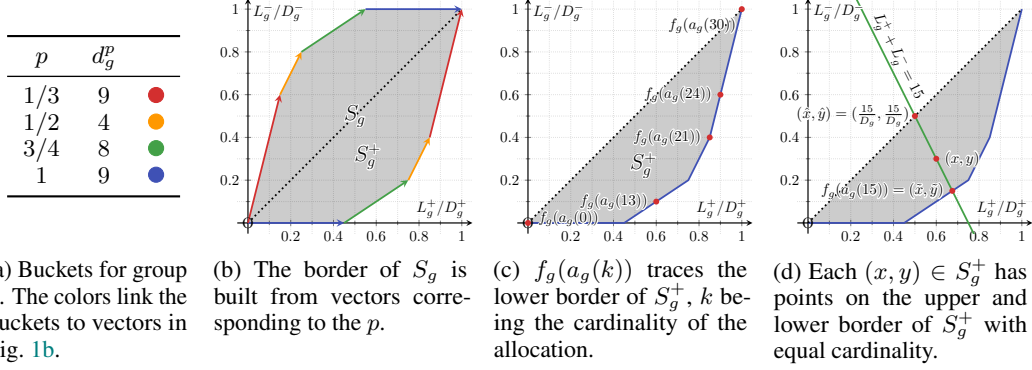

(a) Buckets for group $g$. The colors link the buckets to vectors in Fig. 1b.

(b) The border of $S_g$ is built from vectors corresponding to the $p$.

(c) $f_g(a_g(k))$ traces the lower border of $S_g^+$, $k$ being the cardinality of the allocation.

(d) Each $(x,y) \in S_g^+$ has points on the upper and lower border of $S_g^+$ with equal cardinality.

| $p$ | $d_g^p$ | |
|-----|--------|---|
| 1/3 | 9 | ● |
| 1/2 | 4 | ● |
| 3/4 | 8 | ● |
| 1 | 9 | ● |

Figure 1: Example group $g$. $D_g^+ = 20$ and $D_g^- = 10$.

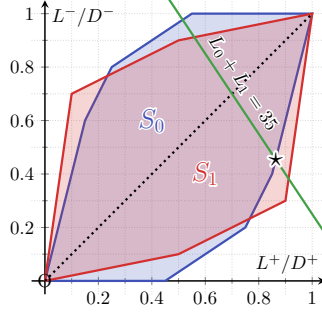

Figure 2: Superimposition of $S_0$ and $S_1$. The cardinality line is drawn in green, the optimal solution is marked by a star.

group specified in Table 1a, the shape of $S_g$ is shown in Fig. 1b. $S_g$ is convex as the image of the convex space of allocations under the linear function $f_g$. Furthermore, the diagonal $(x,x)$ for $0 \leq x \leq 1$ is a subset of $S_g$ as the image of the uniform allocations. We will restrict our investigation to the area $S_g^+$ "right of" that line, i.e., the set $\{(x,y) \in S_g \mid x \geq y\}$, since the arguments for the other half of $S_g$ are symmetric. As the intersection of convex sets, $S_g^+$ is still convex.

Consider a specific value of $p \in P_g$ and the allocation that gives $d_g^p$ units to bucket $p$ and none to all other buckets. Applying $f_g$ to this allocation gives us a vector $v_p := (p\, d_g^p/D_g^+, (1-p)\, d_g^p/D_g^-)$. For example, as described in Table 1a, the bucket $3/4$ has size 8, which means that it contains 6 positive agents and 2 negative agents. In Fig. 1b, this bucket is represented by the green vector $v_{3/4} = (6/20, 2/10)$. Since, in each $v_p$, both components are nonnegative, all these vectors point in a direction between right ($p = 1$) and up ($p = 0$). Because the slope is proportional to $(1-p)/p$, the slope of the $v_p$ decreases monotonically in $p$.[3] As hinted at in Fig. 1b, we want to show that the upper border of $S_g^+$ is the line $(x,x)$, whereas the lower border can be constructed by appending the $v_p$ in order of decreasing $p$. Formally, let $a_g$ be a function from the interval $[0, D_g]$ into the set of allocations. For every $k$, $a_g(k)$ is the unique threshold allocation of cardinality $k$. Thus, $a_g(k)$ determines the smallest $p^* \in P_g$ such that $\sum_{p>p^*} d_g^p \leq k$. Then, $a_g(k)$ sets $\ell_g^p := d_g^p$ for all $p > p^*$, $\ell_g^p := 0$ for all $p < p^*$, and $\ell_g^{p^*} := k - \sum_{p>p^*} d_g^p$. As illustrated in Fig. 1c, $f_g \circ a_g$ walks along the sequence of the $v_p$. This allows us to formally describe the shape of $S_g^+$.

**Theorem 1.** $S_g^+$ *is the convex set whose border is the union of the diagonal line* $\{(x,x) \mid 0 \leq x \leq 1\}$ *and the image of* $f_g \circ a_g$.

*Proof.* Clearly, the image of $f_g \circ a_g$ lies within $S_g = im(f_g)$. Moreover, it intersects the line $(x,x)$ in the points $f_g(a_g(0)) = (0,0)$ and $f_g(a_g(D_g)) = (1,1)$. Since the slope of the vectors increases in their layout from left to right, $im(f_g \circ a_g)$ must lie under $(x,x)$, just like a function with increasing slope is convex. Thus, $im(f_g \circ a_g) \subseteq S_g^+$. Because of the rising slopes of the lower

border and the previous observations, the closed curve induced by walking counter-clockwise along $im(f_g \circ a_g) \cup \{(x, x) \mid 0 \le x \le 1\}$ only has left turns. Thus, the interior of the curve is convex.

It remains to show that the convex hull of $im(f_g \circ a_g) \cup \{(x, x) \mid 0 \le x \le 1\}$ encompasses $S_g^+$. Indeed, let $(x, y) \in S_g^+$ be given, and let the allocation $(\ell_g^p)_p$ be a preimage under $f_g$. By assumption, $x \ge y$, and we may assume without loss of generality that $x > y$. Let $(\hat{\ell}_g^p)_p$ be the uniform allocation that sets $\hat{\ell}_g^p := L_g / D_g \, d_g^p$ for all $p$. Clearly, this allocation is mapped by $f_g$ to the signature $(\hat{x}, \hat{y}) := (L_g / D_g, L_g / D_g)$. Finally, let $(\tilde{\ell}_g^p)_p$ be the allocation produced by $a_g(L_g)$ and let $(\tilde{x}, \tilde{y})$ be its image under $f_g$.

As Fig. 1d shows, the images under $f_g$ of all three allocations lie on a line because they all satisfy

$$D_g^+ \, x + D_g^- \, y = L_g. \tag{1}$$

To show that $(x, y)$ lies inside of the convex hull, it is enough to show that it lies in between the two other points. Since $\hat{x} = \hat{y}$ and $x > y$, since both points satisfy Eq. (1), and since $D_g^+$ and $D_g^-$ are positive, we know that $\hat{x} < x$. It remains to show that $\tilde{x} \ge x$. Let $p^*$ be the probability used in the definition of $a_g(L_g)$. If $\tilde{\ell}_g^p = d_g^p$ for all $p > p^*$ and $\tilde{\ell}_g^p = 0$ for all $p < p^*$, the allocations coincide and we are done. Else, by going from $(\ell_g^p)_p$ to $(\tilde{\ell}_g^p)_p$, we just move parts of the allocation from probabilities $p \le p^*$ to probabilities $p \ge p^*$. The cardinality of the allocation must stay the same by Eq. (1). Since, to calculate $L_g^+$, the allocation for every probability $p$ is counted with weight $p$, this moving can only increase $L_g^+$, thus $\tilde{x} \ge x$. Thus, $(x, y)$ lies in the convex hull, and $im(f_g \circ a_g)$ is indeed the lower border of $S_g^+$. □

Let us return to the full setting with multiple groups, and draw the subsets $S_g$ in the same coordinate system, as illustrated in Fig. 2. For any global allocation satisfying equalized odds, the group allocations must be mapped to the same signatures by the corresponding functions $f_g$. Thus, all these allocations must have a signature in the intersection of the $S_g$. Conversely, for any point in the intersection, we can take preimages of that point for each group and obtain an allocation that is well-formed and satisfies equalized odds.

The remaining constraint is the cardinality constraint on the allocation. Any point $(x, y)$ corresponds to allocations that allocate $(\sum_g D_g^+) \, x$ units to positive agents and $(\sum_g D_g^-) \, y$ units to negative agents. Thus, the total cardinality $(\sum_g D_g^+) \, x + (\sum_g D_g^-) \, y$ of such an allocation must equal $k$. This is equivalent to a constraint $y = (k - (\sum_g D_g^+) \, x)/(\sum_g D_g^-)$. Geometrically, this constraint has the shape of a line with negative, finite slope, which we refer to as the *cardinality line* (see Fig. 2). The cardinality line must intersect the line $(x, x)$ at $x = k/(\sum_g D_g)$, and thus intersects $\bigcap_g S_g$ (even $\bigcap_g S_g^+$). This demonstrates that an equalized-odds allocation with the given cardinality always exists.

Note that efficiency, $\sum_g L_g^+$, is proportional to the $x$ coordinate of a point. Thus, efficiency is optimized by selecting an allocation corresponding to the rightmost point in the intersection of the cardinality line and $\bigcap_g S_g^+$. If we trace the lower border of $\bigcap_g S_g^+$, i.e., we keep following the uppermost of the lower borders of the $S_g$, we obtain a convex monotone *maximum curve*. The signature of the most efficient allocation is then simply defined by the intersection of the cardinality line and this curve.[4] This description directly translates into a polynomial-time algorithm.

## 4 Combining Equalized Odds with Fair Division Axioms

We investigate the compatibility between equalized odds and the three fair division axioms formally introduced in Section 2.3. All four properties can be satisfied simultaneously by allocating uniformly across all groups. Thus, the compatibility must be measured in terms of how much efficiency must be sacrificed to simultaneously guarantee the properties.

This is particularly interesting since, if we do not insist on equalized odds, the most efficient allocation algorithm (which simply allocates to buckets in order of decreasing $p$) immediately satisfies resource monotonicity, consistency, and population monotonicity. Thus, the fair division axioms in question do not have an inherent cost to efficiency, in contrast to punctual axioms in related settings [4, 9]. However, two of them will drastically lower efficiency when imposed in addition to equalized odds with respect to the optimal equalized-odds allocation as a baseline.

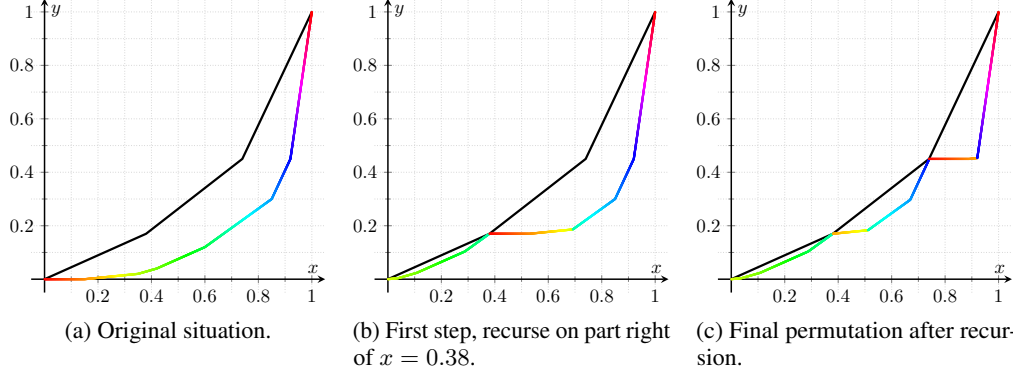

(a) Original situation.  (b) First step, recurse on part right of $x = 0.38$.  (c) Final permutation after recursion.

Figure 3: Illustration for the proof of Theorem 2. The maximum curve is black, and the lower border of $S_g$ is colored, which allows to track how the curve is permuted.

## 4.1 Resource Monotonicity

Fortunately, we can find equalized-odds allocations that satisfy resource monotonicity for free, i.e., while retaining maximum efficiency.

**Theorem 2.** *There is an allocation algorithm that satisfies equalized odds and resource monotonicity, which, on every input, leads to maximum efficiency among all equalized-odds algorithms.*

*Proof sketch.* We sketch the argument here, and relegate the formal proof to Appendix A of the supplementary material. As we described in Section 3, the signature of the optimal equalized-odds allocation is defined by the intersection of the cardinality line and the maximum curve. Increasing the allocation cardinality shifts the cardinality line to a parallel position further to the right. Since the maximum curve is monotone increasing, and since the cardinality line has negative slope, this will shift the intersection further to the right on the maximum curve. This implies that the average allocations to either type cannot decrease, but we need to ensure that the allocation inside of each group does not reduce the allocation to any single bucket. This does not hold for most natural ways of implementing the signature.

It suffices to focus on a single group. We need to associate points on the maximum curve with group allocations of matching signature such that the allocation to any bucket increases monotonically along the curve. It is sufficient to do so for the corners of the maximum curve; convex combinations of the corner allocations directly implement the signatures of a line segment while preserving monotonicity. Geometrically, we can specify such group allocations as a permutation of the $f_g \circ a_g$ curve, where permutation means that we cut the curve into finitely many segments, reorder them, and translate them to form a single connected curve. For example, the colored curves in Figs. 3b and 3c are permutations of the one in Fig. 3a. The permuted curve should touch all corners of the maximum curve. Then, at a corner of the maximum curve, allocate to each bucket $p$ its demand multiplied by the fraction of line segments with corresponding slope that appear before the vertex on the permuted curve. This ensures that the allocation implements the desired signature, and that the allocations increase bucket-wise between corners.

In Lemma 5 in Appendix A of the supplementary material, we describe a recursive algorithm that produces such a reordering. Figure 3 demonstrates this algorithm on an example. In every recursion step, it finds a section of the lower curve that matches the first line segment of the maximum curve, swaps this segment to the left, and then recurses on the subcurves to the right of the intersection. The middle section can be found efficiently without resorting to numerical search; an implementation of the algorithm is included in our code at `https://github.com/pgoelz/equalized`. □

## 4.2 Consistency

Unfortunately, the situation is less rosy for consistency: The only allocation rule that satisfies both consistency and equalized odds is the uniform allocation.

**Theorem 3.** *Let $\mathcal{A}$ be an algorithm that guarantees equalized odds and consistency. Then, $\mathcal{A}$ will allocate uniformly on any given instance.*

*Proof.* We refer to the given instance as Instance I. Obtain Instance II by adding two agents with probability label $1/2$ to each group and by setting the new cardinality constraint to $k\,(n + 2\,\#\text{groups})/n$, such that the average allocation per agent remains the same. Now, every group contains positive and negative agents, and the average allocations $\rho_g^+ := L_g^+/D_g^+$ and $\rho_g^- := L_g^-/D_g^-$ exist. By equalized odds, all $\rho_g^+$ equal a single constant $\rho^+$, and all $\rho_g^-$ equal a single constant $\rho^-$.

Fix any bucket $(g, p)$ with a probability label $p > 0$. We want to show that this bucket will be allocated $\rho^+\,d_g^p$ units in Instance II: Construct an Instance $\text{III}_{g,p}$ from II by removing all buckets except for $(g, p)$ from $g$, along with their allocations. By consistency, this does not change the allocation to any other group; thus, the $\rho^+$ of the other groups are unchanged. Because $(g, p)$ is now the only partially positive bucket, $\rho_g^+$ is just the per-agent allocation of $(g, p)$. By equalized odds, $(g, p)$ is allocated $\rho^+\,d_g^p$ units in $\text{III}_{g,p}$. By consistency, $(g, p)$ receives the same amount in Instance II. Symmetrically, any bucket $(g, p)$ with probability $p < 1$ is allocated $\rho^-\,d_g^p$ units in Instance II.

In any given group $g$, fix the bucket with label $1/2$ and let their common allocation be $\ell_g^{1/2}$. Since $0 < 1/2 < 1$, by the above, $\rho^+ = \ell_g^{1/2}/d_g^{1/2} = \rho^-$. It follows that every single bucket $(g, p)$ in Instance II is allocated $\rho^+\,d_g^p = \rho^-\,d_g^p$ units, so the allocation is uniform. If we remove the inserted agents along with their allocation, we recover Instance I with the original budget $k$. By consistency, the allocation in Instance I was uniform. $\qquad\square$

Intuitively, the incompatibility between equalized odds and consistency is not particularly surprising. By nature, equalized odds is sensitive to the composition of the total application pool, whereas consistency rules out such dependencies. For example, if we remove applicants from one group along with their allocations, this likely changes the mean allocation to positive and negative agents in that group. As a result, the classification on the remaining agents must adapt to still satisfy equalized odds.

### 4.3 Population Monotonicity

For population monotonicity, the situation is also fairly bad, albeit less so than for consistency. In the following theorem, whose proof we defer to Appendix B.1 of the supplementary material, we show that any algorithm satisfying population monotonicity and equalized odds will, on certain inputs, incur arbitrarily high loss in efficiency over the optimum equalized-odds allocation.

**Theorem 4.** *Let $\mathcal{A}$ denote an allocation algorithm satisfying equalized odds and population monotonicity. Then, $\mathcal{A}$ does not give a constant-factor approximation to the efficiency of the optimal equalized-odds algorithm.*

Let us compare this result with Theorem 3, whose assertion holds for any instance. By contrast, Theorem 4 is a worst-case result, and so it leaves room for algorithms satisfying population monotonicity and equalized odds that are significantly more efficient than a uniform allocation in practice. In fact, in Appendix B.2 of the supplementary material, we do construct a non-uniform algorithm with these axiomatic properties that (slightly) outperforms uniform allocations. However, we will shortly see that, on a real dataset, requiring population monotonicity and equalized odds *inevitably* leads to efficiency close to uniform allocations.

## 5 Empirical Results

We evaluate our approach on a dataset relating the FICO credit scores of $174\,047$ individuals to credit delinquency. The dataset is based on TransUnion's TransRisk scores, and was originally published by the Federal Reserve [2]. We use a cleaned and aggregated version made publicly available by Barocas et al. [3] at `https://github.com/fairmlbook/fairmlbook.github.io/tree/master/code/creditscore`. For each of four races (white, black, Hispanic, Asian), the individuals are partitioned into buckets for 198 credit score values. For each bucket, we can compute its size and fraction of non-defaulters. Our code is publicly available at `https://github.com/pgoelz/equalized`.

For different numbers $k$ of loans to be given out, Fig. 4 shows the efficiency loss entailed by insisting on certain fairness properties. As a baseline, we use the optimal non-fair allocation that greedily allocates to agents in descending order of $p$, regardless of their race. Insisting on equalized odds — and, by Theorem 2, even additionally insisting on resource monotonicity — only incurs a small efficiency penalty of less than 3.5%. Even uniform allocation loses at most 30% efficiency since 70%

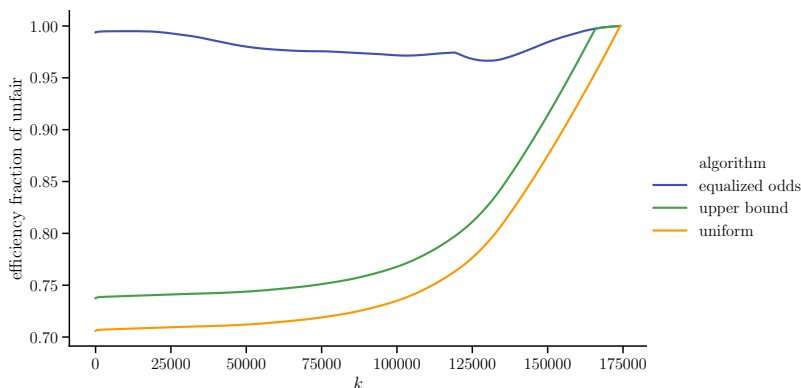

Figure 4: Efficiency of three different equalized-odds algorithms on the FICO dataset, as a function of $k$ and as a fraction of the optimal allocation without fairness constraints.

of agents in the dataset do not become delinquent. The higher $k$ becomes, the more even the optimal non-fair algorithm is forced to allocate to agents that might default, and the lower the relative loss of uniform allocation becomes. Nevertheless, as long as $k$ is not a large fraction of the number of agents, we suspect the price of consistency to be unacceptably high — as is evident from the fact that banks use credit scoring at all.

The most interesting line is the third algorithm. Since we do not have a characterization of the best algorithms satisfying equalized odds and population monotonicity, we test an algorithm that, on every instance, will be at least as efficient as every such algorithm. This algorithm is based on the observation that, if we remove all buckets from a group except for one with probability in $(0, 1)$, any algorithm satisfying equalized odds must give this bucket its proportional share of $k$ in the resulting instance. If population monotonicity is satisfied, this gives us an upper bound on the allocation to the bucket in the original instance. By maximizing for efficiency subject to these constraints and equalized odds with a linear program, we obtain the desired upper bound on every equalized-odds algorithm that satisfies population monotonicity. As the graph shows, insisting on population monotonicity forces us into an efficiency dynamic that is essentially that of uniform allocation. While there is a gap of a few percentage points between the two curves, part of it might be explained by the looseness of our upper bound. Just as in the case of consistency, population monotonicity seems to be unacceptably costly unless we can satisfy a large fraction of the demand.

## 6   Discussion

We have shown that equalized odds in a setting with cardinality-constrained resources is perfectly compatible with the classic fair division axiom of resource monotonicity. However, our theoretical and empirical results imply that equalized odds is grossly incompatible with consistency and (more importantly) population monotonicity.

Why is that a problem? On a practical level, the paradoxes these axioms are meant to prevent can lead to real difficulties. For example, as mentioned in Section 2.3, a violation of population monotonicity may give rise to a situation where we need to decrease a student's financial aid because another student *declined* to accept aid. On a conceptual level, it is hard to justify and explain the design of allocation algorithms that behave in such counter-intuitive ways.

In summary, our results tease out new tradeoffs between notions of fairness. We also believe our work strengthens the case against equalized odds as a tenable standard for fair machine learning.

## Acknowledgments

This work was partially supported by the National Science Foundation under grants IIS-1350598, IIS-1714140, CCF-1525932, and CCF-1733556; by the Office of Naval Research under grants N00014-16-1-3075 and N00014-17-1-2428; by a J.P. Morgan AI Research Award; and by a Guggenheim Fellowship.

## Footnotes

[1] In a recent paper [1], Balcan et al. argue for envy-freeness as a new notion of individual fairness for classification when preferences are heterogeneous.

[2] If a group possesses only positive or only negative agents, all average allocations for its type are possible and it imposes no constraint on the other type. We will thus set $S_g := [0,1]^2$ in these cases.

[3] For $p = 0$, we consider the slope to be infinite.

[4]This point is unique because of the possible slopes for the cardinality line and the line segments making up the maximum curve.

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
