[Supplementary Material]

# Appendix

## A  Formal Proof of Theorem 2

Recall that our goal is to define an allocation mechanism that, given a signature of the optimal equalized-odds allocation, produces an allocation that satisfies this signature and ensures that the allocation to any bucket increases when additional resources are added.

Let us first formally define the maximum curve $m$. For every value $k$, we intersect the corresponding cardinality line with the lower borders of all $S_g$. We select the leftmost of these intersection signatures to be $m(k)$. Note that, at any point, the curve follows the shape of the lower border of some $S_g$, and that it may only change the $S_g$ it traces at intersection points between these lower borders. It follows that the curve is still a finite polygon chain, that its tangential angle exists except for finitely many exceptions, that this angle is between $0$ (right) and $\pi/2$ (up) where it exists, and that the angle increases along the curve where it exists. For every cardinality constraint $k$, the signature produced by the optimal equalized-odds algorithm is exactly $m(k)$.

For each group $g$, we would like to implement the signature $m(k)$, for any given $k$, such that resource monotonicity is not violated inside of this group. Formally, we are looking for a monotone function $j_g : [0, D] \to \prod_{p \in P_g} [0, d_g^p]$ such that $f_g(j_g(k)) = m(k)$ for all $k$. Note that it suffices to define such a function only for the $k$ that correspond to corners of $m$, as we can interpolate between these corners to obtain well-behaved solutions for other values of $k$. Indeed, if the graph of $m(\theta\, k_1 + (1 - \theta)\, k_2) = \theta\, m(k_1) + (1 - \theta)\, m(k_2)$ for two such values of $k$ and all $0 \le \theta \le 1$, setting $j_g(\theta\, k_1 + (1 - \theta)\, k_2) = \theta\, j_g(k_1) + (1 - \theta)\, j_g(k_2)$ will inherit monotonicity and $f_g(j_g(k)) = m(k)$ if these properties hold for $k_1$ and $k_2$.

We will define such $j_g$ by *reordering* the curve $a_g$, which we define later. First, note that $a_g$ is (component-wise) Lipschitz-continuous, since an increase in $k$ by $\epsilon$ will change each agent's allocation by at most $\epsilon$. Thus, $a_g$ is absolutely continuous, and it holds that $a_g(k) = \int_0^k a_g'(k)\, dk$ (a Lebesgue integral) for all $k$, where the derivative $a_g'$ exists everywhere except for finitely many exceptions. We call a function $r : [0, D_g) \to [0, D_g)$ a reordering for $g$ if it is a bijection and if there exist $0 = p_0 < p_1 < \cdots < p_n = D_g$ such that, for all $i \in \{0, 1, \ldots, n - 1\}$ and $x \in [p_i, p_{i+1})$, it holds that $r(x) = r(p_i) + (x - p_i)$. Intuitively, $r$ is a reordering of a partition of $[0, D_g)$ into finitely many subintervals. We say that $r$ induces a *permutation* of $a_g$, which is the function $r[a_g] : [0, D_g] \to \prod_{p \in P_g} [0, d_g^p]$, where $r[a_g](k) = \int_0^k a_g'(r(k))\, dk$ for all $k < D_g$, and $r[a_g](D_g) = (d_g^p)_p$. This function is absolutely continuous on $[0, D_g)$ and remains so through the addition of the point $D_g$ with its left limit. Since $r[a_g]$ is obtained by integrating nonnegative values from $a_g'$, it is still monotone.

Let $k_1 < k_2 < \cdots < k_n$ be the values of $k$ corresponding to corner points of $m$. By Lemma 5, we can find a reordering $r$ and $s_1 < s_2 < \cdots < s_n$ such that $f_g(r[a_g](s_i)) = m(k_i)$ for all $i$. If, for all groups $g$, we set $j_g(k_i) := r[a_g](s_i)$ for all $i$ and interpolate linearly between these points, this defines an allocation algorithm that is resource monotone. Moreover, it still satisfies equalized odds without efficiency losses, since the optimal signature is always implemented. $\qquad\square$

**Lemma 5.** *Let $c$ and $d$ be finite polygon chains in $\mathbb{R}^2$, represented as simple curves. Let all their tangential angles lie between $0$ and $\pi/2$, and let these angles increase monotonically along the curves (where defined). Let $c$ and $d$ both start in a common point and end in a common point, and let $d$ lie below $c$. Then, there exists a reordering $r$ of the domain of $d$ such that $r[d]$ visits all corner points of $c$.*

*Proof.* By scaling, we may assume without loss of generality that $c$ and $d$ both have the domain $[0, 1]$, that $c(0) = d(0) = (0, 0)$, and that $c(1) = d(1) = (1, 1)$. We prove the claim by induction on the number of line segments in $c$. The induction step is illustrated in Fig. 3.

If there is only a single line segment, the identity reordering satisfies the claim.

Else, let $k_{seg}$ be the preimage of the first corner of $c$, and let $(\xi_{seg}, v_{seg}) := c(k_{seg})$ be the $x$ and $y$ dimension of this segment. Denote the $x$ and $y$ components of $d$ by $d_x$ and $d_y$, respectively. By our assumption on the angle of $d$, both functions increase monotonically. For any $x$ coordinate $0 \le \xi \le 1$, let $d_x^{-1}(\xi)$ denote the smallest $0 \le k \le 1$ such that $d_x(k) = \xi$. There is always at least one such $k$

by the intermediate value theorem, and this $k$ is unique for all $\xi < 1$ (there can be multiple $k$ on a final, upward-facing line segment).

Define a function $h : [d_x^{-1}(\xi_{seg}), 1] \to \mathbb{R}_{\geq 0}$ by setting $h(k) := d_y(k) - d_y(d_x^{-1}(d_x(k) - \xi_{seg}))$. Geometrically, $h$ slides a window of width $\xi_{seg}$ over the graph of $d$ and measures the growth of the curve in $y$ direction along this window. Since, by assumption, $d$ lies below $c$, we know that $h(d_x^{-1}(\xi_{seg})) \leq \upsilon_{seg}$. At the same time, the average slope of both $c$ and $d$ is 1. The slope of $c$'s first line segment can be at most that, since the slope increases along the curve. Similarly, the average slope of the window measured by $h(1)$ must be at least 1.

It follows that $h(1) \geq \upsilon_{seg}$. Since $h$ is continuous, by the intermediate value theorem, there is some $k_{right}$ such that $h(k_{right}) = \upsilon_{seg}$. If we set $k_{left} := d_x^{-1}(d_x(k) - \xi_{seg})$, we know that $d_x(k_{right}) - d_x(k_{left}) = \xi_{seg}$ and $d_y(k_{right}) - d_y(k_{left}) = \upsilon_{seg}$.

Define a reordering $r' : [0, 1) \to [0, 1)$ by setting

$$r'(k) = \begin{cases} k_{left} + k & k \in [0, k_{right} - k_{left}) \\ k - (k_{right} - k_{left}) & k \in [k_{right} - k_{left}, k_{right}) \ , \\ k & k \in [k_{right}, 1) \end{cases}$$

i.e., by swapping the intervals $[0, k_{left})$ and $[k_{left}, k_{right})$. It must hold that $r'[d](k_{right} - k_{left}) = (\xi_{seg}, \upsilon_{seg})$.

Now concentrate on the restriction of $r'[d]$ to the interval $[k_{right} - k_{left}, 1]$. It is still a polygon chain, and, since its tangential angles all come from $d$, they lie between 0 and $\pi/2$. Since we only took out a middle segment in the succession of angles, the angles still increase monotonically along the curves. Restrict $c$ to $[k_{seg}, 1]$. Then, the two curves have a common starting point $(\xi_{seg}, \upsilon_{seg})$ and endpoint $(1, 1)$. Finally, the restriction of $r'[d]$ will still lie below the restriction of $c$ because the only changed part took its derivatives from a prefix of $d$, which used to fit below the flattest stretch of $c$, so it will now fit under a steeper stretch of $c$. These observations allow us to apply the induction hypothesis, and obtain a reordering $r''$ (without the scalings described at the beginning of this proof). Define a new reordering $r$ to be equal to $r'$ on $[0, k_{right} - k_{left})$, and to equal $r' \circ r''$ on the remaining interval. This leaves us with a reordering such that the graph of $r[d]$ visits all corners of $c$. $\qquad\square$

## B  Equalized Odds and Population Monotonicity

### B.1  Inapproximability

**Theorem 4.** *Let $\mathcal{A}$ denote an allocation algorithm satisfying equalized odds and population monotonicity. Then, $\mathcal{A}$ does not give a constant-factor approximation to the efficiency of the optimal equalized-odds algorithm.*

*Proof.* Let $a$ be a large integer, to be chosen later. Let Instance I contain two groups, 0 and 1. Group 0 contains a bucket labeled $\frac{a-1}{a}$ with $a$ many agents and a bucket labeled 0 with $2a$ agents. Group 1 contains a bucket labeled 1 with a single agent and a bucket with $2a^2 - a - 1$ many agents labeled 0. Set $k := 2a$.

What efficiency can the optimal equalized-odds algorithm obtain in this instance? Since Group 1 is perfectly classified, the algorithm's behavior is determined by the intersection of the cardinality line and the lower border of $S_0$. The cardinality line is determined by $2a = ax + (1 + 2a + 2a^2 - a - 1)y = ax + (2a^2 + a)y$. The first segment of the border is induced by threshold allocations that only give to the first bucket of Group 0. If we allocate $0 \leq t \leq a$ units to this bucket, we get $x = t/a$ and $y = t/(a(2a+1))$. Plugging these equations into each other, we obtain an intersection at $x = 1$ and $t = a$. This $t$ is in the permissible bounds for the first segment, which means that we indeed have found the intersection of cardinality line and lower border. The optimal equalized-odds algorithm will achieve a total efficiency of $ax = a$, $a - 1$ of which will be obtained from Group 0.

Now consider Instance II, in which Group 1 remains the same but we remove the bucket with the label 0 from Group 0. Since Group 0 contains a single bucket (labeled with probability $(a-1)/a \notin \{0, 1\}$), its convex shape is exactly the diagonal line. Thus, the mean allocation for positive agents must equal the mean allocation for all agents, i.e., $2a/(2a^2 + 2a) = 1/(a+1)$. All positive agents together receive $a/(a+1)$ units.

Assume that $\mathcal{A}$ guarantees an $\alpha$-approximation of the efficiency obtained by the optimal equalized-odds algorithm, where $\alpha > 0$. Choose $a \geq \alpha^{-1}$. Then, in Instance I, $\mathcal{A}$ must allocate at least $\alpha\,a \geq 1$ units to the positive agents. However, if we remove the 0 bucket from Group 0, we obtain Instance II, in which the same positive agents receive strictly less than one unit of the good. This contradicts population monotonicity. $\qquad\square$

## B.2 Non-Uniform Algorithm Satisfying Equalized Odds and Population Monotonicity

**Proposition 6.** *There exists an allocation algorithm that satisfies equalized odds and population monotonicity and that dominates uniform allocation in terms of achieved efficiency.*

*Proof.* We obtain these properties by maximizing for efficiency subject to equalized odds and the constraint that every agent's allocation must lie in $[k/(n + \frac{1}{2}), k/(n - \frac{1}{2})]$. Let's take our usual diagram. We definitely give $k/(n + \frac{1}{2})$ to every agent, so we know that we start at the point $(k/(n + \frac{1}{2}), k/(n + \frac{1}{2}))$. From here on, the situation is very similar to the original "most efficient equalized-odds" one, just that we shrink each agent to accept at most $(k/(n - \frac{1}{2}) - k/(n + \frac{1}{2}))$ additional units of the good instead of one unit.[5] Again, this gives us a convex set of implementable signatures that starts at $(k/(n + \frac{1}{2}), k/(n + \frac{1}{2}))$ and ends at $(k/(n - \frac{1}{2}), k/(n - \frac{1}{2}))$. We know that the cardinality line crosses the intersection of these spaces, because it must run through the point $(k/n, k/n)$. If we take the point where the cardinality line crosses the border of the intersection of convex sets, this defines our allocation.

This algorithm satisfies equalized odds, since we still select a single signature for all groups from the diagram. Why does it satisfy population monotonicity? Have a Instance I, and get a Instance II by adding additional agents to I. $k$ is the same between both instances; let the number of agents be denoted by $n_{\mathrm{I}}$ and $n_{\mathrm{II}}$, respectively. By assumption, $n_{\mathrm{II}} \geq n_{\mathrm{I}} + 1$. If we run the algorithm on both instances, we are guaranteed that every agent in Instance I receives at least $k/(n_{\mathrm{I}} + \frac{1}{2})$ units. If we run it on Instance II, every agent receives at most $k/(n_{\mathrm{II}} - \frac{1}{2}) \leq k/((n_{\mathrm{I}} + 1) - \frac{1}{2}) = k/(n_{\mathrm{I}} + \frac{1}{2})$ units. Thus, no agent can receive more in Instance II than in Instance I; population monotonicity must hold. $\qquad\square$

## Footnotes

[5]To be precise, $min(k/(n - \frac{1}{2}), 1) - k/(n + \frac{1}{2})$ in case $n - \frac{1}{2} < k$.