[Reviews · NeurIPS 2019]

Reviewer 1



Originality: somehow high since those concepts have not been analyzed before together. Quality: The claims are correct, they formalize somewhat a limitation of the equalized odds (it cannot provide partition independence, it can behave strangely as new points are added). Clarity: The paper is clear, with the exception that the motivation or implications (beyond theoretical curiosity) might be simply hard to comprehend. Significance: On the one hand, it's good to clarify but one could argue that the premise on testing equalized odds (or equal opportunity, or statistical parity) on those axioms was doomed to begin with. Partition independence (refered to as consistency here) states that you could divide people and run the algorithm among those newly formed groups separately and come to the same conclusion. But, precisely, equalized odds is meant to be sensitive to the composition of the applicants pool. For hiring or school admission, if you transform arbitrarily the applicant pools, say by removing a bunch of high-quality or low-quality from a protected group, that certainly has to transform the classification for equalized odds. Unless a form of base rate imbalance is maintained, it does seem impossible to reconcile with that axiom outside of corner cases. The violation of population monotonicity is more surprising. Consider an admission process is not just merit based but corrected for fairness for equal odds (for instance because of some bias in the data). The addition of a high applicant from a protected group will certainly improve the odds of protected group, hence remove the pressure not to give the resource among other applicants. But it also increases competition, so in that case it does not seem too easy to predict. It's not super clear, however, why not satisfying this particular axiom will bring real concern on that definition. Adding someone in the competition can help you, such as removing someone in the competion can hurt. Would that really bring concern to a metric of fairness that is anyway very new and not terribly satisfying to begin with? I have read the author response and my opinion remains the same.

Reviewer 2



This paper presents and assesses the compatibility of three geometric interpretations of equalized odds and commonly-known existing axioms of fairness within fair division literature: resource monotonicity, consistency, and population monotonicity. Compatibility in this case is measured related to how much efficiency is sacrificed to guarantee the properties in question simultaneously. The paper shows that equalized-odds allocations satisfy resource monotonicity while also retaining maximum efficiency. This is not the case for the other two axioms presented. The authors show that uniform allocation is the only rule for which consistency and equalized odds are both satisfied. Similarly, population monotonicity and equalized odds generally incurs high loss in efficiency over the optimum allocation on inputs. Notably, the findings also show that fair division axioms are not inherently costly for efficiency (as they are with punctual axioms). The paper is significant in that it addresses previously-unaddressed definitions of fairness and attempts to operationalize them empirically. As of yet, there is not much literature relating machine learning with methodology from fair division. The authors provide a fair amount of clarity by providing clear objectives, definitions, and theorems. However, the paper is limited in that cardinality constraints are generalized to binary decisions, which may not always be the case in applied settings. Financial aid or loan settings may qualify various amounts of financial resources to applicants, but the algorithm assumes that allocations are equally spread across members. EDIT: Thanks to the authors for responding to a misunderstanding in my review. After reading the response, my vote to accept remains unchanged.

Reviewer 3



The paper is very well written and I believe is of substantial significance for the emerging field of fairness in machine learning. The axiomatisation of desiderata for machine learning models is a lofty goal, and I believe the field would benefit from more approaches of this flavour. My main criticism is regarding the notation and technical definitions. There are a lot of subscripts and superscripts and used extensively. Likewise there is an extensive set of jargon that seems to be introduced at various points throughout the paper. For example encountering the sentence "...satisfies resource monotonicity if increasing $k$ does not...", the variable k is not defined anywhere in that subsection, or section, and I had to read over from the beginning to find its definition (in the introduction).

[Author Response · NeurIPS 2019]

We thank all the reviewers for the helpful and supportive reviews. The reviewers made a range of suggestions concerning the presentation of the paper, which we take seriously; we will do our best to incorporate them into the revised version of the paper.

Since the reviewers did not ask any (non-rhetorical) questions, we focus in our rebuttal on one potential misunderstanding.

**Reviewer #2:** *". . . the paper is limited in that cardinality constraints are generalized to binary decisions, which may not always be the case in applied settings. Financial aid or loan settings may qualify various amounts of financial resources to applicants, but the algorithm assumes that allocations are equally spread across members."*

Please note that our model already generalizes binary allocations to continuous allocations between 0 and 1 for each agent. The purpose of this generalization is precisely to capture financial aid or loan settings, which provide different amounts of resources to different applicants. This is discussed in the first three paragraphs of Section 2.1.

We do assume that all agents *within the same bucket* (e.g., those that have the same probability of repaying a loan) receive the same allocation. As we describe in Section 2.1, this is necessary to guarantee equalized odds without further information (except for extremal values of $p$, where it is not interesting). This assumption also seems natural from an ethical viewpoint. We emphasize that agents in different buckets may receive different allocations.

[Meta-Review · NeurIPS 2019]

The authors show a connection between statistical metrics of fairness and the standard axioms in the fair division literature. This is an interesting and timely piece of work.